# Perception and knowledge of the effect of climate change on infectious diseases within the general public: A multinational cross-sectional survey-based study

Max van Wijk[1,2☯], SoeYu Naing[1,2☯], Silvia Diaz Franchy[1], Rhiannon T. Heslop[1], Ignacio Novoa Lozano[1], Jordi Vila[2,3], Clara Ballesté-Delpierre[2]*

1 Departament de Sanitat i Anatomia Animal, Universitat Autònoma de Barcelona (UAB), Cerdanyola del Vallès, Spain, 2 ISGlobal, Hospital Clínic—Universitat de Barcelona, Barcelona, Spain, 3 Department of Clinical Microbiology, Centre for Biomedical Diagnosis, Hospital Clínic, Barcelona, Spain

☯ These authors contributed equally to this work.
* clara.balleste@isglobal.org

**Data Availability Statement:** All relevant data are within the manuscript and its Supporting Information files.

## Abstract

Infectious diseases are emerging and re-emerging due to climate change. Understanding how climate variability affects the transmission of infectious diseases is important for both researchers and the general public. Yet, the widespread knowledge of the general public on this matter is unknown, and quantitative research is still lacking. A survey was designed to assess the knowledge and perception of 1) infectious diseases, 2) climate change and 3) the effect of climate change on infectious diseases. Participants were recruited via convenience sampling, and an anonymous cross-sectional survey with informed consent was distributed to each participant. Descriptive and inferential analyses were performed primarily focusing on the occupational background as well as nationality of participants. A total of 458 individuals participated in this study, and most participants were originally from Myanmar, the Netherlands, Spain, United Kingdom and the United States. Almost half (44%) had a background in natural sciences and had a higher level of knowledge on infectious diseases compared to participants with non-science background (mean score of 12.5 and 11.2 out of 20, respectively). The knowledge of the effect of climate change on infectious diseases was also significantly different between participants with and without a background in natural sciences (13.1 and 11.8 out of 20, respectively). The level of knowledge on various topics was highly correlated with nationality but not associated with age. The general population demonstrated a high awareness and strong knowledge of climate change regardless of their background in natural sciences. This study exposes a knowledge gap in the general public regarding the effect of climate change on infectious diseases, and highlights that different levels of knowledge are observed in groups with differing occupations and nationalities. These results may help to develop awareness interventions for the general public.

**Funding:** We acknowledge support from the Spanish Ministry of Science and Innovation through the "Centro de Excelencia Severo Ochoa 2019-2023" Program (CEX2018-000806-S), and support from the Generalitat de Catalunya through the CERCA Program. This study was performed by students of the Erasmus Mundus Joint Master Degree (EMJMD) in Infectious Diseases and One Health (IDOH+) and initiated by the Universitat de Autònoma in Barcelona, Spain. The Master program is funded by the European Commission and M.V.W. and SY.N. are Erasmus+ scholarship holders. The European Commission is not involved in any manner in this study. The funders had no role in study design, data collection and analysis, decision to publish, or preparation of the manuscript.

**Competing interests:** The authors have declared that no competing interests exist.

## Introduction

Climate change remains one of the major One Health issues, and so are infectious diseases. The transmission of certain infectious diseases has already been altered due to processes driven by climate and environment abnormalities [1]. Vector-borne disease outbreaks are expected to occur in the current temperate regions as a consequence of increasing temperatures while altered patterns of rainfall and floods will also alter the vector species presence, density, fitness, and eventually the transmission dynamics [2]. As an example, dengue and chikungunya outbreaks in southern Europe have been gradually increasing in recent years [3]. Even a single extreme weather event such as the 2015–2016 El Niño phenomenon has contributed to many outbreaks of infectious pathogens including water and vector-borne diseases such as cholera, West Nile virus and Lyme diseases throughout the world [4]. Even though the awareness of climate change has largely increased in recent years, the knowledge gap of the effect of climate change on infectious diseases remains unchanged [5].

It is important to get an insight into the awareness of this topic among the general population and measure the knowledge level based on demographic variables to determine the underlying knowledge gaps in each area. Despite the strong scientific evidence on the link between climate change and infectious diseases, there is a lack of research that assesses the awareness and knowledge on this topic in both the general population and healthcare professionals [6]. One of the only perception and knowledge assessments studies done up to date was conducted in China among healthcare professionals working at the Chinese Center for Disease Control (CCDC). Only one third of the CCDC staff showed to have great knowledge that climate change has an impact on infectious diseases, highlighting the need for training and educational programs on this matter [7].

The current study is the first multinational cross-sectional research that aimed to evaluate the knowledge, perception and attitude on the impact of climate change on the dissemination of infectious diseases within the general public. The study focused on the three main topics, namely 1) infectious diseases, 2) climate change and 3) the effect of climate change on infectious diseases. The main objective of the study was to determine if participants with a background in natural science have a greater knowledge of these topics than participants with other backgrounds. This study also performed in-depth analyses of the existing interventions or public awareness programs with a focus on climate change and infectious diseases, and evaluated if the level of knowledge varies based on the geographical regions.

## Methodology

### Design of survey

A survey was designed to assess the knowledge and perception of climate change, infectious diseases and the effect of climate change on infectious diseases of every participant (S1 File). A cover page was included to inform the participant with the aim of the study and to obtain consent. The survey was composed of 21 questions in total. The demographic characteristics of participants were assessed in the first questions and included age, nationality, country of residence, educational level, occupational status, background in natural sciences, and travel history (the number of continents visited, if the participants have lived abroad, and the number of countries visited in the last five years). All questions and answer options were written in English. The answer options for the educational level and the animal species mentioned along the questions were accompanied with Dutch and Spanish translations, as a significant number of surveys were distributed to citizens of the Netherlands and Spain.

**Perception assessment.** One question composed of 19 items captured the participant's opinion on infectious diseases, climate change and the link between these two topics. The

answer options were according to a five-point Likert scale (strongly disagree, disagree, neither agree nor disagree, agree and strongly agree) along with the *don't know* option.

**Knowledge assessment.**   Participants were asked to identify infectious diseases out of a list of diseases, risk factors of getting infectious diseases, and animals that transmit infectious diseases. Next, assessment of the participant's knowledge on infectious diseases was evaluated by 6 statements on this particular topic. Another question composed of 15 statements was used to assess the knowledge on climate change. The last knowledge assessment contained 14 statements and focused on the effect of climate change on infectious diseases. All these knowledge assessments had three answer options: true, false and *don't know*.

**Attitude assessment.**   The attitude assessment included questions asking for the interest in the topic before and after the survey, the primary way of information consumption and the willingness to learn more about the effect of climate change on infectious diseases.

## Ethical statement

The survey was designed to collect opinion and knowledge data on the relationship between climate change and infectious diseases. According to the current regulation, this survey did not require the approval of an Ethics Committee on Clinical Investigation, as human subjects research was not conducted. No personal data, including special categories of data, was collected, meaning that only fully anonymized was processed.

Prior to participation in this study, participants were asked to read and approve the following consent statement: "This research study, conducted by students of the master program Infectious Diseases and One Health, is designed to help our understanding of climate change and infectious diseases. On the following pages you'll be asked to fill in a short questionnaire. All responses that you provide in this study are kept strictly confidential. Your participation is voluntary and you may discontinue participation at any time. Participation involves no more than minimal risk". Participants were able to withdraw their consent at any time and contact information of one of the authors (M.V.W. or SY.N.) was provided to allow participants to address any concern.

Participants were given informed consent including the purpose of the study, what they were being asked to do at the beginning of the survey. Participants were informed that this study is completely voluntary, has no risk involved, and the data obtained are protected confidentially with a unique study-ID number. All participants read the description and gave the consent to participate in the study by responding "proceed". Participants were also informed that they could withdraw their consent at any time.

## Data collection

Questionnaires were generated and collected using an online platform—SoGoSurvey [8] and the survey was opened for participants between March 21th and March 25th 2020. Participants were recruited via convenience sampling methods using several social media platforms. The sequence of sub-items per question was randomized for every participant to prevent order bias. All questions were mandatory to be answered before submitting the survey. Only completed questionnaires were saved and collected in the utilized platform.

## Data transformation and grading

After all surveys were collected, two new variables were added in the dataset: continent of nationality and residence. These variables were assigned based on countries of nationality and residence after data collection by the first two authors (M.V.W. and S.Y.N. independently used a two-letter coding system and the data transformation was later compared to check for

manual editing errors). For analysis purposes, individual responses of all knowledge assessments were re-coded as 1 (for correct answers), -0.5 (for incorrect answers) and 0 (for *don't know* option). Afterwards, the scores for every knowledge assessment were normalized to a maximum score of 20 and every participant was scored according to the percentage of the maximum score (scores 85% and higher = A, 70 to 85% = B, 55 to 70% = C, 40 to 55% = D and scores 40% and lower = F). The answer keys of the questions are compiled in the S2 File and the scoring key is annexed in S3 File. Every participant received three scores for the knowledge assessments: one for infectious diseases, one for climate change and one for the effect of climate change on infectious diseases. The Likert scale of the perception assessment was transformed into a simpler scale: agree, neutral, disagree and don't know.

## Visualization and statistics

R-studio version 1.1.447 was used for the visualization of all data. Several statistical tests were used to examine statistically significant differences between the sample subsets. Independent t-tests were performed to compare scores between participants with a background in natural sciences and other backgrounds. Linear regression was used to determine if age was correlated with individual scores. One-way ANOVA was run to compare scores of participants with different nationalities.

## Results

### Participants characteristics

In total, 458 unique responses were collected and 35.2% (n = 161) were originally from an English-speaking country (United Kingdom, United States, Canada, Australia or New Zealand). Regarding gender, 38.7% (n = 177) were males, 60.9% (n = 279) were females, and 0.4% (n = 2) did not enclose their gender. The median and mean age were 27 and 32.8 respectively, with all ages ranging from 18 to 78 years. The majority of participants' nationality was European (50.7%, n = 232), followed by Asian (28.4%, n = 130) and North American (13.8%, n = 63). A distribution of nationalities is visualized in Fig 1. Regarding the occupational background, the vast majority of our participants were either full-time employed (50.9%, n = 233)

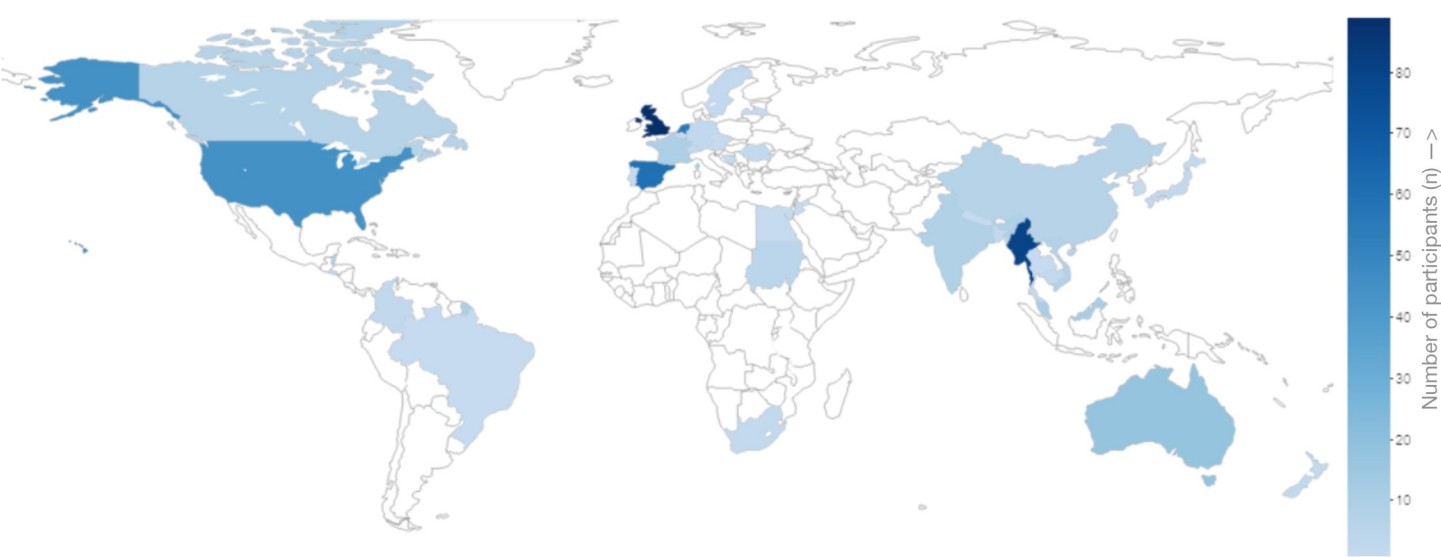

**Fig 1. Visualization of the distribution of nationalities.** Countries are color coded according to the proportional number of participants.

or students (27.5%, n = 126). Almost half of the sample population (44.3%, n = 203) either studied or worked in the field of natural sciences, while the rest (55.7%, n = 255) had other backgrounds. Most participants had a Bachelor's degree or were enrolled in a Bachelor's program during the time this survey was conducted (50.0%, n = 229), followed by a Master's degree (28.6%, n = 131). More than half (59.4%, n = 272) have lived abroad, of which 77.9% (n = 212) have lived on another continent than where they were originally from. A more detailed description of the obtained demographic characteristics is found in Table 1.

## Perception assessment

**Infectious diseases.** Almost half of the participants (44.1%) considered themselves as someone who knows more about infectious diseases than the general population. This was respectively 69.5% and 23.9% amongst participants with and without a background in natural sciences.

In general, the vast majority (64.6%) was afraid of getting an infectious disease. European participants were the least afraid (51.7%), compared to North American nationals (71.4%) and participants of Asian origin (87.7%). Regarding travel prophylaxis, the vast majority (70.5%) checked if they need vaccination prior to traveling to a tropical country.

In line with this observation, more than half (56.1%) was more afraid of getting an infectious disease when traveling to a tropical region although differences were also detected based on the nationality: in this case, participants of European nationality were more afraid (72.0%), compared to North American nationals (41.3%) and participants with an Asian origin (37.7%).

In terms of identifying infectious diseases, the majority was able to identify malaria (90.4%), HIV (89.7%), tuberculosis (89.1%) and Lyme disease (64.2%) in the category of infectious diseases. A minority of the sample wrongly considered allergies (7.0%), asthma (4.2%), diabetes (2.0%) and obesity (1.8%) as belonging to this group.

When asking for the sources of getting an infectious disease, no proper hand hygiene (94.6%) was considered to be the top risk factor, followed by drinking unclean water (90.0%) and having sexual intercourse (87.8%). Breastfeeding was only considered as a risk factor by a minority (28.8%). A small proportion (15.9%) identified smoking as a risk factor for getting an infectious disease.

Regarding the knowledge of the participants referring to treatment of infectious diseases and how to prevent them, almost one out of five (19.7%) thought that antibiotics can be used to treat viral infections, while one out of three (30.1%) did not think that vaccines can be used to prevent bacterial infections.

**Climate change.** Almost half of our participants (47.3%) considered themselves as someone who knows more about climate change than the general population. This was respectively 52.2% and 47.1% for participants with and without a background in natural sciences. And the vast majority (81.9%) knew the definition of the greenhouse effect. Almost all participants (92.1%) believed that humans are responsible for global warming and a great proportion (92.1%) reported that climate change will be more severe in the future. With respect to behavioral change, 58.7% agreed and 26.9% strongly agreed with the statement that they have changed their habits in recent years to minimize the impact on the environment. A majority (85.8%) believed that climate change will negatively impact our accessibility to food and a similar proportion (85.8%) thought that intensive farming contributes to climate change, while a minority (22.1%) believed that eating vegetables and fruits does not contribute to climate change.

**Effect of climate change on infectious diseases.** In a third part of the survey, both concepts (climate change and infectious diseases) were combined in order to study the current

**Table 1. Overview of obtained demographic characteristics of participants.**

| | Total | Natural Sciences | Other backgrounds | P-value |
|---|---|---|---|---|
| | *n = 458* | *n = 203* | *n = 255* | |
| **Age** | | | | *0.005* |
| Median (IQR) | 27 (25–37) | 26 (24–33) | 27 (25–40) | |
| **Gender** | | | | *0.58* |
| Female | 279 (60.92%) | 129 (63.55%) | 150 (58.82%) | |
| Male | 177 (38.65%) | 73 (35.96%) | 104 (40.78%) | |
| Prefer not to answer | 2 (0.44%) | 1 (0.49%) | 1 (0.39%) | |
| **Continent of Nationality** | | | | *0.16* |
| Africa | 8 (1.75%) | 6 (2.96%) | 2 (0.78%) | |
| Asia | 130 (28.38%) | 63 (31.03%) | 67 (26.27%) | |
| Europe | 232 (50.66%) | 97 (47.78%) | 135 (52.94%) | |
| North America | 63 (13.76%) | 30 (14.78%) | 33 (12.94%) | |
| Oceania | 20 (4.37%) | 5 (2.46%) | 15 (5.88%) | |
| South America | 5 (1.09%) | 2 (0.99%) | 3 (1.18%) | |
| **Employment** | | | | *< 0.0001* |
| Student | 126 (27.51%) | 76 (37.44%) | 50 (19.61%) | |
| Part-time employed | 48 (10.48%) | 21 (10.34%) | 27 (10.59%) | |
| Full-time employed | 233 (50.87%) | 94 (46.31%) | 139 (54.51%) | |
| Unemployed / seeking for opportunities | 18 (3.93%) | 3 (1.48%) | 15 (5.88%) | |
| Retired | 30 (6.55%) | 7 (3.45%) | 23 (9.02%) | |
| Prefer not to answer | 3 (0.66%) | 2 (0.99%) | 1 (0.39%) | |
| **Level of Education** | | | | *< 0.0001* |
| High school | 34 (7.42%) | 6 (2.96%) | 28 (10.98%) | |
| Vocational degree | 23 (5.02%) | 7 (3.45%) | 16 (6.27%) | |
| Bachelor's degree | 229 (50.00%) | 107 (52.71%) | 122 (47.84%) | |
| Master's degree | 131 (28.60%) | 54 (26.60%) | 77 (30.20%) | |
| Doctoral degree | 41 (8.95%) | 29 (14.29%) | 12 (4.71%) | |
| **Lived abroad** | | | | *0.75* |
| No | 186 (40.61%) | 85 (41.87%) | 101 (39.61%) | |
| Yes—on the same continent | 60 (13.10%) | 24 (11.82%) | 36 (14.12%) | |
| Yes—on another continent | 212 (46.29%) | 94 (46.31%) | 118 (46.27%) | |
| **Number of Countries Visited (last 5 years)** | | | | *0.20* |
| 0 countries | 24 (5.24%) | 14 (6.90%) | 10 (3.92%) | |
| 1–3 countries | 140 (30.57%) | 70 (34.48%) | 70 (27.45%) | |
| 4–6 countries | 146 (31.88%) | 57 (28.08%) | 89 (34.90%) | |
| 7–10 countries | 85 (18.56%) | 34 (16.75%) | 51 (20.00%) | |
| > 10 countries | 63 (13.76%) | 28 (13.79%) | 35 (13.73%) | |

The demographic information is shown in absolute numbers and percentage of the (sub)sample. Apart from the complete sample, the demographics are displayed for participants with a background in natural sciences and other backgrounds. The latter two are compared on statistically significant differences using Fisher and Wilcoxon tests.

knowledge of the general population and identify potential knowledge gaps. One out of three participants (30.4%) said that they are well informed about the effect of climate change on infectious diseases. Not even half (40.9%) of participants with a background in natural sciences reported to be well informed. This was even lower for participants with other occupational backgrounds (24.7%).

More than half of the participants (84.7%) believed that global warming has already caused damage to human health, while a minority (28.4%) reported that global warming has not yet affected humans but will do so in the future. When asking for the connection between both concepts, a percentage of 70.3% believed that climate change influences infectious diseases whereas a small minority (6.1%) said that there is no direct link between climate change and infectious diseases.

The majority of participants (75.3%) reported that climate change and extreme weather conditions influence the spread of infectious diseases and a similar proportion (80.8%) reported that the transmission of infectious diseases may be favored by floods, although some (22.9%) thought that climate change can reduce the transmission of infectious diseases.

With respect to the effect of climate change in infectious disease in the future, more than half (69.4%) believed that disease outbreaks will increase in the future because of climate change. Almost all participants (86.7%) knew that mosquitoes survive better at warmer temperatures, and a similar proportion (71.0%) reported that vector-borne diseases such as malaria can become a problem for countries with moderate climates in the future.

## Knowledge assessment

**Infectious diseases.** The mean score of the knowledge assessment on infectious diseases was 11.8 out of 20 (95% CI [11.5;12.0]), and a small proportion had a mark equivalent to grade A or B (Fig 2A). Participants with a background in natural sciences scored significantly higher (mean score of 12.5, 95% CI [12.2;12.8]) than participants with other backgrounds (mean scores of 11.2, 95% CI [10.9;11.6]; $p < 0.001$). Participants with a European nationality also scored significantly higher than participants of Asian origin (mean scores 12.0, 95% CI [11.7;12.4] and 11.1, 95% CI [10.7;11.6] respectively; $p = 0.04$). Participants originally from North America had a mean score of 11.9 (95% CI [11.3;12.6]), there was no significant difference between European and North American nationalities ($p = 1.0$), and between North American and Asian nationalities ($p = 0.33$). There was no statistically significant difference in mean score between post-secondary, undergraduate and graduate degree holders except that the score of vocational degree holders was significantly lower than that of doctoral degree holders ($p = 0.01$) No association was found for other continents of origin ($p > 0.13$) or age ($R = 0.003$, $p = 0.14$).

**Climate change.** The mean score of all participants was 14.4 out of 20 (95% CI [14.1;14.7]) and more than half (65.7%) had a mark equivalent to grade A or B (Fig 2B). Participants with a North American nationality (mean score of 14.7, 95% CI [14.0;15.4]) had the same mean score as participants with a European nationality (mean score of 14.7, 95% CI [14.3;15.2]). Participants with an Asian nationality yielded a lower score (mean score of 13.6, 95% CI [13.1;14.1]). The difference in mean score was statistically significant between participants with European and Asian nationalities ($p < 0.01$), whereas this was not the case between Asian and North-American nationals ($p = 0.09$). Level of education was positively associated with level of knowledge, and the difference in mean score of post-secondary, undergraduate students and graduate students was statistically significant ($p<0.001$). There was no statistically significant difference in mean score depending on background in natural sciences ($p = 0.32$) or age ($R = 0.005$, $p = 0.07$).

**Effect of climate change on infectious diseases.** Overall, a moderate proportion (40.2%) had sufficient knowledge of the impact of climate change on infectious diseases (equivalent to grade A and B, see Fig 2C). The mean score was 11.8 (95% CI [11.3;12.2]), and the difference was statistically significant between participants with (mean score of 13.1, 95% CI [12.5;13.8]) and without background (mean score of 10.9, 95% CI [10.3;11.5]) in natural sciences

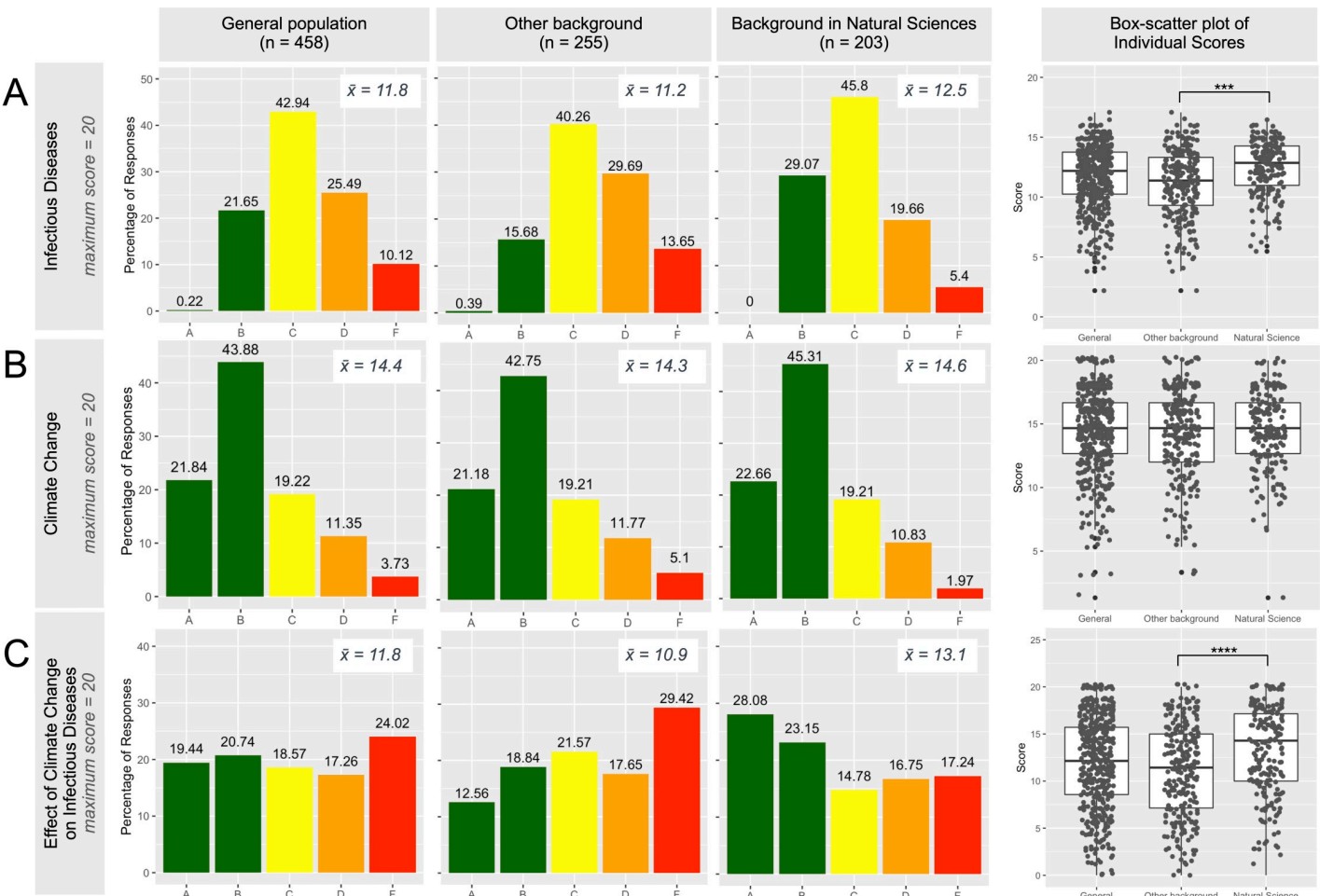

**Fig 2.** Overview of grouped scores on the three topics assessed in the survey: (A, top panel) infectious diseases, (B, middle panel) climate change and (C, lower panel) the effect of climate change on infectious diseases. Results are given for the total sample (n = 458), participants without (n = 255) and with (n = 203) background in natural sciences. Average scores are included in every plot (indicated by $\bar{x} =$). Scores are according to the percentage of the maximum score (85% and higher = A, 70 to 85% = B, 55 to 70% = C, 40 to 55% = D and scores 40% and lower = F). Box-scatter plot of individuals scores for the three subsets are visualized on the right hand side. Significance as *** at p-value < 0.001 and **** p-value < 0.0001.

(p < 0.0001). Level of education was not statistically significant with level of knowledge (p>0.40). There was no significant difference in knowledge between participants with European, Asian or North-American nationality (p > 0.14) and the score was not correlated with age (R = -0.00001, p = 0.32).

**Top five represented nationalities.** The following 5 nationalities were the most represented in our sample (72.3%, 331 out of 458): United Kingdom (n = 89), Myanmar (n = 80), Spain (n = 60), Netherlands and United States (n = 45). An overview and comparison of demographic variables per nationality are displayed in S4 File.

Regarding the knowledge assessment on climate change, participants with a British nationality scored significantly higher than participants with a Burmese (p < 0.01) or Dutch nationality (p = 0.02). No statistically significant differences were present between the other analyzed nationalities (p > 0.23) (see Table 2). Regarding the knowledge of infectious diseases, the scores were identical for all analyzed nationalities (p > 0.07). In terms of knowledge of climate change and infectious diseases, participants that were originally from Spain yielded a significantly higher score than participants from Myanmar (p < 0.001), the Netherlands (p = 0.02)

**Table 2. Overview of scores for the top five represented nationalities.**

|  | Climate change | Infectious diseases | Climate change and infectious diseases (interaction) |
|---|---|---|---|
| **Myanmar (MM)** | **13.60 ± 0.32** | **10.96 ± 0.30** | **10.70 ± 0.55** |
| The Netherlands (NL) | 13.56 ± 0.53 | 11.60 ± 0.35 | 11.15 ± 0.70 |
| Spain (ES) | 14.57 ± 0.39 | 12.13 ± 0.37 | 13.94 ± 0.49 |
| United Kingdom (GB) | 15.24 ± 0.34 | 11.95 ± 0.27 | 11.05 ± 0.53 |
| United States (US) | 14.83 ± 0.38 | 12.25 ± 0.37 | 12.41 ± 0.86 |

Myanmar, the Netherlands, Spain, United Kingdom and the United States were the most represented nationalities in the study dataset. The table contains the mean scores ± standard error (SE) per analyzed nationality.

or the United Kingdom (p < 0.01). There was no statistically significant difference between participants originally from Spain and the United States (p = 0.34).

## Attitude assessment

In general, nearly half of our participants (49.8%) had not previously considered the effect of climate change on infectious diseases. This was lower for participants with a background in natural sciences (38.4%) compared to participants with other occupational backgrounds (59.2%).

Following completion of the survey, more than half (57.4%) reported that it changed their opinion about the topic, and a vast majority (75.8%) would like to learn more about the effect of climate change on infectious diseases. Among participants with a background in natural sciences, this survey has still changed the opinion of almost half (47.3%) and the majority want to learn more about this topic (80.8%). Although many participants with other backgrounds reported that the survey has changed their opinion (65.5%), they are less willing to learn more about this (71.8%).

Participants reported that they have mainly obtained information about this topic via the internet and social media (33.8%), television programs and documentaries (20.1%) and educational programs and conferences (19.7%). For participants with a background in natural sciences, less than half (36.0%) reported that educational programs and conferences are the main source of information.

## Discussion

The study findings showed that participants with a background in natural sciences had a greater knowledge about infectious diseases and understood that infectious diseases are sensitive to climate change. Their mean scores were significantly higher than participants with other backgrounds. However, the level of knowledge on climate change was not associated with having an occupational background in natural sciences or not. The general public demonstrated to have a high knowledge on climate change, and this was also reflected by the self-assessment of the participants. These findings were similar to a previous study from Yale University that assessed the American knowledge solely on climate change in 2010 [9]. In contrast to the present findings, a knowledge assessment study conducted among American college students reported that students with a science background showed stronger knowledge on climate change than non-science students [10]. The discrepancy in these findings could be due to the difficulty of questions, topics addressed, as well as the length of the survey. Also, general public knowledge on climate change might have improved over time since communication on climate change has been intensified in recent years [18].

The current findings showed that the awareness of climate change was more pronounced in the general public compared to that of infectious diseases, likely associated with an extended media broadcasting for climate change than for infectious diseases [19]. Western nationalities tended to have a higher knowledge on climate change and infectious diseases than oriental nationalities, but not on the link between climate change and infectious diseases. This difference could be explained by the fact that people perceived climate and infectious diseases differently based on their geographical and cultural characteristics. There are more awareness campaigns on climate change and sustainability in Western countries whereas infectious diseases such as water and mosquito-borne diseases are more commonly seen in Asian countries [11, 12].Previous studies have also shown that knowledge and perception were highly variable between communities and countries. A cross sectional knowledge survey in Yemen showed that almost all households (90%) were aware of the signs and symptoms of dengue while only a small proportion (19%) of Bangladeshi households had sufficient knowledge on this infectious disease [13, 14].

The results obtained in the present study clearly indicated that the general public are not fully aware of the role of climate change on infectious diseases. To our knowledge, this is the first multinational study to highlight the knowledge gap of the effect of climate change on infectious diseases under a cross-sectional setting. The CDC study from China in 2016 showed that the vast majority agreed that climate change would affect human health, and weather abnormalities would influence infectious diseases [7]. Contradictory to our results, the majority lacked knowledge on the effect of climate change on infectious diseases. Another study also conducted in China in 2018 showed high knowledge and perception of medical, public health and nursing students on the adverse consequences of climate change on human health. Interestingly, the study results also showed a strong association between the level of knowledge on the causes of climate change and awareness of the effect of climate change on human health [15]. Similar research was conducted among Indonesian adolescents in senior high school. Their findings showed the lack of awareness and knowledge on climate change and health [16]. Again, adolescents had a superficial understanding that climate change has an impact on human health, despite the lack of knowledge on the causes of climate change. Clearly, there is a lack of knowledge and awareness on the consequences of climate change on infectious diseases in different populations. These studies together with the present work highlight the need for integrated educational programs and additional training among healthcare professionals and students to develop a comprehensive global view on the health impact of climate change.

## Limitations and future implications

The use of convenience sampling was a major limitation in this work as it could have imposed selection bias. Fluency in English of participants was not accessed, which might have also imposed bias on their understanding and interpretation. This limitation might have led to the underrepresentation of a part of the population in certain geographical regions. Some questions were excluded from data analysis since they imposed ambiguity and misinterpretation, which could have been omitted by performing a pilot survey. One must also be aware that COVID-19 pandemic might have influenced the knowledge, attitude and perception of the participants since the study was conducted in late March, which was the initial phase of the coronavirus pandemic. Occupation data of the participants was not collected in this survey, but might have been an interesting factor to integrate.

This study has some strengths including the well- collected data that allows for a future in-depth analysis on the association between knowledge and other demographics, such as travel history and whether participants have lived abroad. It has previously been shown that

international travelers are not well informed on infectious diseases at their travel destinations [17]. These findings suggested that perception and knowledge on infectious diseases is not associated with travel behavior, which could be a confounding factor for the knowledge on climate change and infectious diseases as well.

**National policy and awareness interventions.** The current findings have shown that the knowledge levels varied based on geographical locations, the nationality of participants in particular. The differences in awareness implementation strategies on climate change and infectious diseases on a national level might be an explanatory factor for the differences in knowledge of the analyzed nationalities.

Within the European Union, Spain is the first country to have a national climate adaptation plan and strategies in place since 2006, and other European countries followed later (2013 for the United Kingdom and 2012 for the Netherlands) [18]. The Netherlands has thoroughly investigated the effect of climate change on its country from 2007 to 2014 [19] in which a national strategy plan was designed and implemented since 2012 and has been revised in 2016 [20]. This has led to an increased availability of financial resources, but the utilization of these funding and content of awareness campaigns are primarily decentralized [21]. Even though national plans and strategies have been implemented in European countries, the awareness campaigns on the impact of climate change on infectious diseases are rarely occurring on the community level, and the integration of this topic into educational programs is not yet fully developed [20, 22–24]. All Member States of the European Union are addressing climate change and the effect on human health in their policies more than before [18]. The exact content of this public health perspective is highly dependent on the geographical location of the country, and several trends in the policies are clearly visible. Countries with higher latitudes in Europe (e.g. the Netherlands and United Kingdom) are focusing more on air pollution due to climate change, adverse health effects due to extreme weather conditions and diseases contributable to a warmer climate (such as skin cancer and heat shock). European countries that are situated at lower latitudes (e.g. Spain) are more likely to suffer from the introduction of infectious diseases due to change in weather patterns in the near future. This is also reflected in the national policies, as the effect of climate change on infectious diseases and public health is more prevalent in the Spanish strategy plans. High awareness of Spanish citizens combined with the relatively long presence of these action plans might explain why Spanish nationals demonstrated greater knowledge on the effect of climate change on infectious diseases than Dutch and British citizens.

In Asia, Myanmar is considered as one of the top countries that has been most affected by extreme weather conditions in the recent years [25]. Although Myanmar has a similar policy and strategy plan as Spain [26], participants originally from Myanmar obtained a significantly lower score than Spanish citizens. This knowledge difference might be due to differences in socioeconomic status (including level of education), and it is notable that participants with a Burmese nationality were mostly living abroad during the time the study was conducted (S4 File).

In North America, the United States has recently implemented a crosscutting group on climate change and human health (CCHHG). Their objectives include addressing the issue of climate-sensitive infectious diseases to protect public health and strengthen national security [27]. Their action plans emphasize on public engagement for the issues of climate change and infectious diseases. The US government meets this goal by encouraging students and scientists to participate in global scientific activities [28] Moreover, the American institutions are offering and developing a variety of online resources that are easily approachable and accessible for the general public. These might be explainable factors that participants originally from the United States yielded a fairly high score for all three assessed topics.

Overall, it is worth-noticing that all countries are lacking public engagement, effective campaigns and adaptation of the education curriculum that addresses the topic of climate change and infectious diseases despite the political efforts in place.

## Conclusions

Overall, the current study shows that the general population has similar levels of knowledge on climate change while they lack knowledge on infectious diseases and the effect of climate change on infectious diseases compared to participants with a background in natural sciences. Despite the increased effort on promoting climate change awareness, this study highlights the need to implement the educational training and public awareness interventions that address the effect on climate change on emerging and re-emerging infectious diseases.

## Supporting information

**S1 File. Survey.**
(DOCX)

**S2 File. Answer key.**
(DOCX)

**S3 File. Scoring key.**
(DOCX)

**S4 File. Demographic overview of the top 5 represented countries.**
(DOCX)

## Acknowledgments

M.V.W., SY.N., S.D.F., I.N.L. and R.T.H. are students of the Erasmus Mundus Joint Master Degree (EMJMD) in Infectious Diseases and One Health (IDOH+) and part of a consortium composed of Université de Tours (formerly Université François Rabelais) in France, Universitat Autònoma de Barcelona (UAB) in Spain and the University of Edinburgh (UoE) in the United Kingdom. We thank Laila Darwich, Enric Mateu and Marga Martín of the UAB for initiating this project. We would also like to thank Nisha Tucker for her contribution to this study.

## Author Contributions

**Conceptualization:** Max van Wijk, SoeYu Naing, Silvia Diaz Franchy, Rhiannon T. Heslop, Ignacio Novoa Lozano, Clara Ballesté-Delpierre.

**Data curation:** Max van Wijk, SoeYu Naing.

**Formal analysis:** Max van Wijk, SoeYu Naing.

**Investigation:** Max van Wijk, SoeYu Naing, Silvia Diaz Franchy, Rhiannon T. Heslop, Ignacio Novoa Lozano.

**Methodology:** Max van Wijk, SoeYu Naing, Silvia Diaz Franchy, Rhiannon T. Heslop, Ignacio Novoa Lozano.

**Project administration:** Max van Wijk, SoeYu Naing.

**Writing – original draft:** Max van Wijk, SoeYu Naing, Silvia Diaz Franchy, Rhiannon T. Heslop, Ignacio Novoa Lozano.

**Writing – review & editing:** Max van Wijk, SoeYu Naing, Jordi Vila, Clara Ballesté-Delpierre.

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
