## [Decision Letter · Decision Letter 0]

9 Sep 2020

PONE-D-20-20967

Perception and knowledge of the effect of climate change on infectious diseases within the general public: a multinational cross-sectional survey-based study.

PLOS ONE

Dear Dr. Ballesté-Delpierre,

Thank you for submitting your manuscript to PLOS ONE. After careful consideration, we feel that it has merit but does not fully meet PLOS ONE’s publication criteria as it currently stands. Therefore, we invite you to submit a revised version of the manuscript that addresses the points raised during the review process.

We look forward to receiving your revised manuscript.

Kind regards,

Tauqeer Hussain Mallhi, Ph.D

Academic Editor

PLOS ONE

Journal Requirements:

Reviewers' comments:

Reviewer's Responses to Questions

**Comments to the Author**

1. Is the manuscript technically sound, and do the data support the conclusions?

Reviewer #1: Yes

Reviewer #2: Yes

2. Has the statistical analysis been performed appropriately and rigorously? 

Reviewer #1: Yes

Reviewer #2: N/A

3. Have the authors made all data underlying the findings in their manuscript fully available?

Reviewer #1: Yes

Reviewer #2: Yes

4. Is the manuscript presented in an intelligible fashion and written in standard English?

Reviewer #1: Yes

Reviewer #2: Yes

5. Review Comments to the Author

Reviewer #1: Van Wijk and colleagues have performed an interesting survey-based study to investigate the knowledge of the participants on 1) climate change, 2) infectious disease and 3) the effect of climate change on infectious disease. Participants were split primarily in to the general public with and without a background in natural science and by nationality.

Minor points:

It would be interesting to further dissect those participants that did not have a background in natural science based on their occupation. i.e did those participants with higher skilled jobs have a better understanding? This may further reflect the education of the participants.

Further, it would be interesting to investigate the level of education (which it was highlighted that these data were collected) on the knowledge of 1) climate change, 2) infectious disease and 3) the effect of climate change on infectious disease. This may identify were best to target education for specific groups to better inform the general public on infectious disease and/or climate change.

Reviewer #2: The manuscript is interesting, the topic is actual and relevant – infectious diseases, climate changes and the interactions between both, within a general public.

Being a multinational cross sectional survey-based study it allows comparisons among results from different places and persons.

The questionnaire adopted is adequate, well written, concerning:

- personal data (Q1-Q10)

- knowledge in infectious diseases (Q12- Q16.5)

- knowledge on climate change (Q16.1 – Q16.15)

- knowledge on the effect of climate change on infectious disease (Q17.1-Q17.14)

The population studied is diverse, the cohort size adequate.

The data transformation and grading is adequate and allows an easy data interpretation.

The results are well presented and the discussion is easy to follow

Weakness of the work are identified, and I would say that discussion should be slight shortened (6 pages)

6. PLOS authors have the option to publish the peer review history of their article (what does this mean?). If published, this will include your full peer review and any attached files.

Reviewer #1: No

Reviewer #2: No

---

## [Author Response · Author response to Decision Letter 0]

21 Sep 2020

The mansucript has been edited and information added and modified according to the reviewers comments.

The 3 requested files have been submitted:

- Manuscript

- Manuscript with track changes

- Response to reviewers

---

## [Decision Letter · Decision Letter 1]

19 Oct 2020

Perception and knowledge of the effect of climate change on infectious diseases within the general public: a multinational cross-sectional survey-based study.

PONE-D-20-20967R1

Dear Dr. Ballesté-Delpierre,

We’re pleased to inform you that your manuscript has been judged scientifically suitable for publication and will be formally accepted for publication once it meets all outstanding technical requirements.

Kind regards,

Tauqeer Hussain Mallhi, Ph.D

Academic Editor

PLOS ONE

Additional Editor Comments (optional):

Reviewers' comments:

Reviewer's Responses to Questions

**Comments to the Author**

1. If the authors have adequately addressed your comments raised in a previous round of review and you feel that this manuscript is now acceptable for publication, you may indicate that here to bypass the “Comments to the Author” section, enter your conflict of interest statement in the “Confidential to Editor” section, and submit your "Accept" recommendation.

Reviewer #1: All comments have been addressed

Reviewer #2: All comments have been addressed

2. Is the manuscript technically sound, and do the data support the conclusions?

Reviewer #1: Yes

Reviewer #2: (No Response)

3. Has the statistical analysis been performed appropriately and rigorously? 

Reviewer #1: Yes

Reviewer #2: (No Response)

4. Have the authors made all data underlying the findings in their manuscript fully available?

Reviewer #1: Yes

Reviewer #2: (No Response)

5. Is the manuscript presented in an intelligible fashion and written in standard English?

Reviewer #1: Yes

Reviewer #2: (No Response)

6. Review Comments to the Author

Reviewer #1: (No Response)

Reviewer #2: (No Response)

7. PLOS authors have the option to publish the peer review history of their article (what does this mean?). If published, this will include your full peer review and any attached files.

Reviewer #1: No

Reviewer #2: No

---

## [Editor Report · Acceptance letter]

27 Oct 2020

PONE-D-20-20967R1 

Perception and knowledge of the effect of climate change on infectious diseases within the general public: a multinational cross-sectional survey-based study. 

Dear Dr. Ballesté-Delpierre:

I'm pleased to inform you that your manuscript has been deemed suitable for publication in PLOS ONE. Congratulations! Your manuscript is now with our production department. 

Kind regards, 

on behalf of

Dr. Tauqeer Hussain Mallhi 

Academic Editor

PLOS ONE